# Promising Adjuvants and Platforms for Influenza Vaccine Development

**DOI:** 10.3390/pharmaceutics13010068

**Published:** 2021-01-07

**Authors:** Wandi Zhu, Chunhong Dong, Lai Wei, Bao-Zhong Wang

**Affiliations:** Center for Inflammation, Immunity & Infection, Institute for Biomedical Sciences, Georgia State University, Atlanta, GA 30303, USA; wzhu3@gsu.edu (W.Z.); cdong@gsu.edu (C.D.); lwei11@student.gsu.edu (L.W.)

**Keywords:** influenza vaccine, adjuvants, nanoparticles

## Abstract

Influenza is one of the major threats to public health. Current influenza vaccines cannot provide effective protection against drifted or shifted influenza strains. Researchers have considered two important strategies to develop novel influenza vaccines with improved immunogenicity and broader protective efficacy. One is applying fewer variable viral antigens, such as the haemagglutinin stalk domain. The other is including adjuvants in vaccine formulations. Adjuvants are promising and helpful boosters to promote more rapid and stronger immune responses with a dose-sparing effect. However, few adjuvants are currently licensed for human influenza vaccines, although many potential candidates are in different trials. While many advantages have been observed using adjuvants in influenza vaccine formulations, an improved understanding of the mechanisms underlying viral infection and vaccination-induced immune responses will help to develop new adjuvant candidates. In this review, we summarize the works related to adjuvants in influenza vaccine research that have been used in our studies and other laboratories. The review will provide perspectives for the utilization of adjuvants in developing next-generation and universal influenza vaccines.

## 1. Introduction

Influenza epidemics are a severe public health issue each year. During the 2019–2020 influenza season in the United States, about 38 million illnesses, 400,000 hospitalizations, and 22,000 deaths were associated with influenza [1]. Compared with previous flu seasons, higher hospitalization rates were observed among children under four and adults between 18 and 49 [1]. Besides, domestic or wild zoonotic influenza viruses may break the host barriers, jump to humans, and cause influenza pandemics. Frequent human infection with H5N1 and H7N9 in recent years foreshadows emerging pandemics. As it takes time to develop immunity for an emerging virus, an influenza pandemic can be a catastrophe for humans [2,3].

Vaccination is an effective method to protect humans from influenza viral infection or alleviate symptoms induced by influenza-associated diseases. Licensed influenza vaccines include inactivated influenza vaccines (IIV), recombinant influenza vaccines (RIV), and live attenuated influenza vaccines (LAIV) [4]. However, the formulations selected in each flu season are primarily dependent on influenza surveillance data because of the continuous mutation of influenza viruses [5]. A mismatch between the influenza vaccine strains and the circulating influenza strains could significantly reduce vaccine effectiveness. A panel of methods was developed for viral isolation, identification, and sequencing, allowing scientists to rapidly identify mutant strains when a mismatch occurs at the early stage of an influenza outbreak. In this circumstance, novel manufacturing methods would be required to produce large-scale of vaccine supply quickly.

With the above challenges to the seasonal influenza vaccine strategy, universal influenza vaccines that elicit comprehensive, long-term, and broad protection are urgently needed. Various approaches are undertaken to realize such influenza vaccines. The two primary distinctive methods are: (1) Applying conserved epitopes and domains in place of different influenza strains as vaccine immunogens. The well-known conserved antigens include the head-removed hemagglutinin stalk domain (hrHA), neuraminidase (NA), matrix protein 2 (M2), and T cell epitopes resident in influenza internal proteins (such as nucleoprotein (NP) and matrix protein 1 (M1). Some combinations of these antigens have proven to provide cross-protection against different virus challenges in laboratory animals. (2) Improving immune responses by various complementary adjuvants. Adjuvants are molecules or ingredients that are administered with vaccines to improve immune responses. The early innate immune responses in an infection or vaccination program the dimension and magnitude of antigen-specific immune responses [6]. As triggers of innate immune responses, appropriate adjuvants tailor antigen-specific immune responses for optimal protection and immune memory. Through a deep understanding of the immunological mechanisms underlying natural influenza infection for immunity generation and memory, safe and effective adjuvants will be discovered and applied to develop universal influenza vaccines. This article will review the adjuvants that are under preclinical study and early phases of clinical trials. We will focus on the novel adjuvants that might have potent effects in bridging innate and adaptive immune responses and discuss the delivery strategies and routes used to improve influenza vaccine outcomes.

## 2. Licensed Influenza Adjuvants

Aluminium salts (Alum) are the most widely used adjuvants in human influenza vaccines. Other adjuvants licensed for human influenza vaccines in different areas include oil-in-water emulsions (MF59, AS03, and AF03), virosomes, and heat-labile enterotoxin (LT) [7]. Studies have suggested that Alum functions in several ways, such as helping with antigen uptake [8], the induction of interleukin-1ß release by inflammasomes [9], enhancing antigen presentation, and strengthening the interaction between dendritic cells (DCs) and CD4^+^ T cells [10]. The immunological mechanism underlying the function of Alum is still not entirely understood.

During the 2020–2021 influenza season, both trivalent and quadrivalent FLUAD inactivated influenza vaccines containing the adjuvant MF59 were approved for people 65 and older. MF59 is an oil-in-water emulsion, which works very distinctly from Alum. MF59 injection leads to the release of some specific chemokines and cytokines, like CCL2, CCL3 IL-8, and IL-5 [11,12]. MF59 preferentially induces Th2-biased immune responses [13]. Like MF59, AS03 and AF03 are oil-in-water adjuvants that take effect in the fashion of MF59.

Virosomes are lipid vesicles that incorporate the influenza antigens on the surface and encapsulate an aqueous solution. Influenza virosomes incorporate influenza antigens onto the vesicle surfaces to mimic the physical features of viruses, such as shapes and sizes, which enhances antigen uptake and presentation and the subsequent immune cell activation [14,15]. Inflexal^®^ V is a commercially available virosome-based influenza vaccine [16]. LT was previously licensed as a mucosal adjuvant for the influenza vaccine; however, it was later found that LT induced Bell’s palsy in the recipients [17].

The advantages and disadvantages of Alum [7,18], oil-in-water emulsions [18,19], virosomes [20,21], and LT [22] were listed in Table 1. More adjuvants are being developed and studied to improve influenza vaccine outcomes in preclinical and clinical trials.

## 3. Adjuvants in Immune Responses

The progress in understanding innate immunity and its role in directing adaptive immune responses have provided new thoughts for the next generation of adjuvants [23]. As the early responders in an infection, various cell types—including macrophages, DCs, γδ T cells, and NKT cells—can sense and respond to adjuvants. Pattern recognition receptors (PRRs) are expressed inside or on the surfaces of these innate cells. Transmembrane receptors, like Toll-like receptors (TLRs), and cytoplasmic receptors, like the nucleotide-binding and oligomerization domain (NOD)-like receptors (NLRs), are two kinds of PRRs that have been well studied. PRRs recognize pathogen-associated molecular patterns (PAMPs) and danger-associated molecular patterns (DAMPs) and activate the downstream signaling pathways, resulting in pro-inflammatory cytokine generation modulating the humoral and cellular immune responses. PAMPs, DAMPs, or other innate signaling molecules could be potential adjuvants to guide immune outcomes during immunization [24,25]. Some adjuvants (systems) are summarized in Table 2.

### 3.1. TLR Agonists

The activation of TLR signaling pathways is essential for protection against influenza virus infection [26]. Administration of TLR3, TLR9, TLR7, or TLR7/8 agonists resulted in viral inhibition and improved mouse survival [27]. Recent publications have further demonstrated that the combinations of synthesized TLR4, TLR7, and TLR7/8 ligands were potent adjuvants for recombinant influenza HA vaccines in different animal models [28,29,30,31]. Significantly, the TLR4 and TLR7 ligands—such as MPL/R837, TRAC-478, and 1Z105/1V270—synergistically increased antigen-specific, long-lasting humoral immune responses, Th1 cell-mediated or Th1/Th2-balanced immunity, and protection against homologous, heterologous, and heterosubtypic viral challenges [6,28,30]. Intranasal co-administration of a synthetic TLR3 ligand, poly I:C, with inactivated human, avian, or swine influenza vaccines activated mucosal and systemic humoral responses in mice, ducks, or pigs [32,33,34,35]. Thus, the inclusion of appropriate TLR agonists can alter the directions of the immune response.

Flagellin (FliC) is a natural ligand of TLR5 and has proven to be a potent adjuvant when administered with influenza antigens [36]. Our laboratory has studied FliC as a potent adjuvant by constructing various antigen-FliC formulations [37,38,39,40,41,42,43,44]. Skin immunization with recombinant fusion protein 4M2e-FliC induced strong M2e-specific humoral and mucosal immune responses [37]. We also redesigned the 4M2e-FliC construct to include M2e sequences from different influenza subtypes. We demonstrated that microneedle patch (MNP)-based boosting immunizations with the 4M2e-FliC could rapidly broaden influenza-vaccine-induced immunity [43]. An MNP encapsulating 4M2e-FliC and inactivated influenza vaccines (H1N1 and H3N2) was developed and demonstrated to have antiviral efficacy against reassortant A/Vietnam/1203/2004 H5N1 and A/Shanghai/2013 H7N9 virus challenge. However, a recent clinical trial showed an overproduction of inflammatory molecules from the vaccination of a flagellin-M2e fusion protein vaccine (STF2.4 × M2e) [45]. In the trial, the ratio of flagellin to the antigen was fixed due to the fusion protein state of STF2.4 × M2e. One of our previous studies in guinea pigs demonstrated that 0.5 µg of flagellin-adjuvanted HIV virus-like particles (VLPs) induced significantly higher levels of neutralizing antibody responses than non-adjuvanted VLPs without over-production of inflammatory cytokines [46]. An optimal dose of flagellin is to be studied as a safe adjuvant in influenza vaccine development.

TLR9 agonists—unmethylated CpG oligodeoxynucleotides (CpG ODN or CpG)—are among the most promising adjuvants that could be used in humans. The administration of CpG has induced Th1-biased responses. The inclusion of CpG in the inactivated influenza vaccines enhanced T cell responses and provided protection against a heterosubtypic influenza infection [47]. Two research groups simultaneously reported that combinations of MPL and CpG (MPL + CpG) induced various inflammatory cytokines and chemokines within one day. MPL + CpG double-adjuvanted influenza vaccines improved protective efficacy with elevated IgG2a antibodies and Th1-biased immune response [48,49].

### 3.2. Cytosolic Nucleic Acids

In addition to TLR signaling pathways, detecting and responding to pathogens through nucleic acid sensors is another approach to activating innate immune responses. Various RNA and DNA-sensing receptors, such as RIG-I-like receptors (RLRs) and cyclic GMP-AMP synthase (cGAS), regulate downstream signaling pathways and subsequent cytokine secretion. The activation of RIG-I induced the production of type I interferons (IFNs) and pro-inflammatory cytokines. 5′ triphosphorylated and diphosphorylated short dsRNAs (5′pppRNA), synthesized small molecule compounds, and poly I: C are well-studied RIG-I-associated adjuvants for enhancing the efficacy of influenza vaccines [50,51,52].

The cGAS-STING pathway was activated during an influenza virus infection and played essential roles in the battle against the infection [53]. 2′3′-cyclic GMP-AMP (cGAMP) is a natural agonist of stimulator of interferon genes (STING), which strongly augmented protective cellular and humoral immune responses induced by influenza vaccines. Meanwhile, compared with intramuscular injection, cGAMP showed a superior adjuvant effect on cutaneous vaccination. cGAMP-adjuvanted H5N1 induced long-lasting protective immunity [54]. An important discovery is that lung delivery of pulmonary surfactant (PS)-biomimetic liposomes encapsulating cGAMP-augmented influenza vaccines induced humoral and CD8^+^ T cell immune responses in mice. The immunity conferred strong cross-protection against distant H1N1 and heterosubtypic H3N2, H5N1, and H7N9 viruses for at least 6 months [55,56]. cGAMP was also reported as a mucosal adjuvant by intranasal immunization. Co-delivery of H7N9 vaccines with cGAMP enhanced humoral, cellular, and mucosal immune responses in mice [57]. These studies suggest that cGAMP is a promising adjuvant for developing a universal influenza vaccine.

### 3.3. Agonists for Inflammasomes Activation

Inflammasomes are another essential component of the innate immune system. The three major types are NOD-like receptor protein 3 (NLRP3) inflammasomes, NLR-family CARD domain-containing protein 4 (NLRC4) inflammasomes, and absent in melanoma 2 (AIM2) inflammasomes [58].

Inflammasomes regulate inflammation by activating caspase-1 and releasing pro-inflammatory cytokines such as IL-1β and IL-18 [59]. Aluminium salts [60], MF-59, AS03, QS-21 [61], and chitosan [62] have been shown to activate inflammasome as part of their mechanisms of immunological activities [63]. Flagellin can also activate inflammasomes through its cytosolic receptor NLRC4 [64]. Nucleic acids (DNA and RNA) could be used for both vaccines and potential adjuvants, as both DNA and RNA can activate inflammasomes [65]. Single- and double-stranded RNAs (ssRNAs and dsRNAs) are recognized by RIG-I, which can subsequently activate NLRP3 inflammasome [66]. In contrast, double-stranded DNA (dsDNA) can be sensed by the AIM2 inflammasome signaling pathway [67].

The activation of inflammasomes vitally inhibits influenza virus infection by limiting the lung damage or enhancing adaptive immune responses through the activation of downstream IL-1R signaling events [68,69,70]. While most works have been focused on the role of NLRP3 in adjuvanticity, the characteristics of other NLR-associated inflammasomes are also being investigated, such as the NRLC5 inflammasome [71]. Inflammasome activators could be used as adjuvants to strengthen immune responses.

### 3.4. Activators of Immune Cells

Besides the primary professional antigen-presenting cells (APCs), i.e., DCs and macrophages, γδ T cells, NK cells, NKT cells, neutrophils, eosinophils, and mast cells are essential components of the innate immune system. Synergistically activating the function of different innate immune cells could facilitate comprehensive immune responses and provide broad protection. Molecules participating in the activation of these cells are promising adjuvants.

Invariant (i) NKT cells, a significant subset of NKT cells, serve as a bridge between the innate and adaptive immune responses. Activated iNKT cells rapidly secrete both Th1 and Th2 cytokines to facilitate DCs maturation and germinal center (GC) B cell responses [72]. Glycolipid ligand α-galactosylceramide (α-GalCer) is a stimulator of iNKT cells. The adjuvanticity of α-GalCer has been studied for influenza vaccines in different animal models. α-GalCer is a promising adjuvant for influenza vaccines, which enhances antigen-specific antibody production and increases protective efficacy.

Mast cells are important innate immune cells and play a crucial role in fighting against bacterial and viral infection. Activated mast cells regulate the migration of immune cells and the induction of adaptive immune responses. Mast cells can be stimulated by various activators, including compound 48/80 (C48/80), IL-33, IL-18, alum, and IgG immune complexes [73,74]. Intranasal immunization with C48/80 adjuvanted recombinant influenza HA elicited protective immunity against 2009 pandemic H1N1 influenza in mice [75]. Intranasal vaccination with the IL-18 and IL-33 adjuvanted recombinant influenza vaccine significantly enhanced antigen-specific antibody responses in systemic compartments and mucosal sites and increased mouse survival during lethal influenza challenges [76].

### 3.5. Cytokines and Chemokines

Other cytokines that modulate immune cells are potential adjuvants for influenza vaccines. IL-1β is an inflammatory cytokine released from its proprotein by inflammasome-mediated caspase-1 activation [77]. Mucosal delivery of recombinant adenoviral vectors (rAd) encoding IL-1β enhanced influenza HA-specific antibody responses. rAd-IL-1β-adjuvanted immunization increased mucosal and systemic T cell immune responses, local tissue-resident memory T cell population, and improved protection against heterologous influenza strains H1N1, pH1N1, H3N2, and H7N7 [78].

Tumor necrosis factor (TNF) is one product of C48/80 stimulation, which directs DC migration [74]. A combination of influenza antigens with particulate TNF increased GC activities and mouse survival rates after a lethal influenza challenge [79].

Granulocyte-macrophage colony-stimulating factor (GM-CSF) is an immunomodulatory cytokine that promotes the maturation of granulocytes and macrophages and regulates DC homeostasis [80]. Skin vaccination with GM-CSF-adjuvanted influenza vaccines induced robust long-term antibody responses and improved mouse protection against lethal influenza challenges [81]. GIFT4 is a novel cytokine that was constructed in our lab by fusing GM-CSF and interleukin-4. We found that a glycolipid (GPI)-anchoring GIFT4 enhanced the immunogenicity of HIV VLPs [82].

Chemokines are a group of small chemoattractant proteins that play a critical role in the tissue-directed migration of immune cells. The use of chemokines as adjuvants is a potential option for developing novel influenza vaccines to direct immune effectors to vulnerable sites for intensive protection [83]. Mucosa-associated epithelial chemokine (CCL28) and cutaneous T-cell-attracting chemokine (CCL27) represent attractive homing chemokines. CCL27, CCL28, and their receptor, CCR10, are essential regulators of mucosal immune responses and important for lymphocyte recruitment to specific mucosal sites. Our lab has previously demonstrated that GPI-anchored CCL28 (GPI-CCL28) acted as an effective adjuvant in an influenza VLP vaccine, which induced robust immune responses at systemic and mucosal compartments and provided significant cross-protection against heterologous viral infection [84] (Figure 1B).

## 4. Particulate Adjuvants and Self-Adjuvanted Particulate Vaccine Platforms

Particles of various types have been investigated as vaccine adjuvants for both injection and mucosal routes. Encapsulating antigens into nanoparticles or onto their surfaces has been shown to enhance antigen-specific antibody responses and cell-mediated immunity. Nanoparticles are an important class of nanoscale materials that have been engineered with controllable and tunable physicochemical properties, including size, shape, structure, and surface chemistry.

The development of self-adjuvanted nanoparticle platforms carrying molecular adjuvants and antigens is highly desirable because such particles can efficiently transport antigens to target cells and activate innate signaling. Different self-adjuvanted nanoparticle platforms are displayed in Figure 1.

### 4.1. Gold (Au) Nanoparticles

Gold (Au) nanoparticles are one of the most common inorganic nanoparticles used for vaccine formulations. Due to the strong affinity of thiol moieties with Au nanoparticle surfaces, thiol-modified polymers or biomolecules (proteins, peptides, oligonucleotides, targeting ligands) can be readily conjugated onto the nanoparticles. With good biosafety and biocompatibility, Au nanoparticles have been used for developing influenza and HIV vaccines [42,85,86,87]. We previously developed multifunctionalized dual-linker gold nanoparticles (AuNPs) to co-deliver influenza antigens and FliC [86] (Figure 1A). Compared with soluble proteins, self-adjuvanted AuNPs-HA/FliC enhanced antigen uptake and induced significantly improved antibody responses. We reported later that the AuNP-HA and AuNP-FliC particle mixtures generated strong mucosal and systemic immune responses and protected immunized mice against lethal influenza virus challenges [42]. The self-adjuvanted Au nanoparticle influenza vaccines demonstrated a high potential for an intranasal influenza vaccine with enhanced vaccine efficacy.

### 4.2. Lipid Nanoparticles

Lipid nanoparticles (LNPs), typically composed of an ionizable lipid, cholesterol, lipid conjugated with polyethylene glycol (PEGylated lipid) and a helper lipid, have recently been recognized as a novel delivery system. LNPs have been used for antigen and adjuvant codelivery. CpG-incorporated LNPs improved the adjuvant effects of CpG ODN and broadened the protection against influenza virus infection [88]. Combinations of TLR ligands with lipid formulations are of particular interest. A split influenza vaccine with co-encapsulated TRAC-478 (a synthetic dual TLR adjuvant) liposome delivery system stimulated strong humoral immune responses and induced Th1-cell-mediated immunity; The immunity protected immunized mice against a heterologous influenza challenge [28] (Figure 1E).

mRNA vaccines are a promising alternative to other vaccine approaches. One mRNA vaccine formulation could easily include multiple mRNAs encoding different viral antigens and innate signaling triggers. mRNA LNPs are one of the novel mRNA vaccine technologies. Nucleoside-modified mRNA LNPs have induced increased GC responses. Full-length influenza HA mRNA-encapsulated LNPs induced HA-stalk-specific antibodies that provided cross-protection in mice [89]. Meanwhile, intradermal (ID) delivery of combined influenza HA stalk, neuraminidase (NA), matrix-2 ion channel (M2), and NP mRNA LNPs have induced robust immune responses and provided broad protection [89,90,91,92]. Thus, codelivery of appropriate adjuvants with the mRNA LNPs is an effective method to enhance the immune response.

### 4.3. Protein Nanoparticles

Compared with other particulate platforms, protein nanoparticles are exclusively antigenic and adjuvant proteins. With self-assembling motifs or under some physical condition (like desolvation), proteins can automatically assemble into nanoparticles. With virtually no polymer or nanocarrier, protein nanoparticles have an extremely high antigen-loading capacity. The methods for the preparation and characterization of double-layered protein nanoparticles have been well established in our laboratory. We have found that layered protein nanoparticles composed of an HA stalk from both H1N1 and H3N2 influenza strains and M2e induced immune protection against homo- and heterosubtypic influenza A viruses [93]. This double-layered protein nanoparticle platform can be adapted to accommodate different influenza conserved antigens. We have developed nanoparticles by desolvating M2e or NP into cores and crosslinking HA stalks, HA, NA, or NP on the core surfaces as coating antigens [94,95] (Figure 1C). The immunogenicity and protective efficiency of these nanoparticles have been determined. Based on this nanoplatform, we are interested in incorporating different adjuvants or targeting molecules together with influenza antigenic proteins into nanoparticles to improve the vaccine outcomes. These self-adjuvanted nanoparticles will be fabricated into MNP for skin delivery in our laboratory.

### 4.4. Other Nano-Platforms

Other types of nanoparticles have also been studied to construct self-adjuvanted vaccine formulations, such as silver (Ag) nanoparticles and calcium phosphate (CaP) nanoparticles. In a recent report, Ag nanoparticles demonstrated promising results in boosting the mucosal immunity of inactivated flu vaccines in a pulmonary immunization and protected against lethal influenza infection [96]. The inclusion of silver nanoparticles induced much stronger antigen-specific IgA in bronchus-associated lymphoid tissue (BALT), reducing the lung viral titers and concomitant lung inflammation. Compared with other adjuvants, such as poly I:C and AddaVax, Ag nanoparticles displayed superior potential in providing potent mucosal immunity potency and protecting mice against influenza infection. For example, a single oral immunization of the AgNP/H5 DNA vaccine in chickens successfully induced antigen-specific antibody responses and cell-mediated immune responses, and enhanced cytokine production [97].

Calcium phosphate nanoparticles (CaP) are a kind of biodegradable nanoparticle with excellent biocompatibility. Knuschke et al. reported the high potential of CaP nanoparticles in inducing cellular immunity when formulated with a conserved influenza A/PR/8/34 (H1N1) HA peptide and a TLR9 agonist, CpG [98]. These nanoparticles were efficiently internalized by DCs in vivo and elicited potent T-cell-mediated immunity; Greatly increased numbers of antigen-specific, IFN-γ-producing CD4^+^ and CD8^+^ effector T cells were detected. Moreover, CaP nanoparticles were useful adjuvants in multiple administration routes and powerfully induce the balanced T helper type-1 (Th1) and Th2 immune responses [99,100].

Biodegradable synthetic polymeric (PLGA) nanoparticles containing influenza antigens, TLR4, and TLR7 ligands (MPL + R387) have been reported to induce synergistic increases in antigen-specific antibodies and complete protection against lethal influenza virus strains challenge in mice and rhesus macaques [6] (Figure 1D). The immune-stimulating complex (ISCOM) is another type of particulate adjuvant. It is composed of antigens, cholesterol, phospholipid, and the immunostimulatory saponin. Matrix M was the third generation of ISCOM and used as an adjuvant in clinical trials for influenza vaccines [101,102].

In addition to three-dimensional nanomaterials, two-dimensional sheet-like nanomaterials, such as graphene oxide (GO) nanoparticles, also attract interest in constructing novel self-adjuvanted vaccine platforms. As the typical example, GO nanoparticles demonstrated great potential as vaccine delivery systems, because of their extraordinary advantages, including the high loading capacity resulting from the intrinsically high aspect ratio and ultra-large surface area, the easy and flexible surface modification with the presence of a wealth of chemical groups (carboxylic acid, epoxy and hydroxyl groups, etc.), and the biocompatible and nonimmunogenic features. GO-nanoparticle-based vaccines can be prepared in many ways, including direct absorption via hydrogen bonding, hydrophobic or π–π stacking interactions, and chemical conjugation with the rich chemical groups. Flexible surface modification with polymers makes the design and fabrication of GO vaccine formulations amiable.

Inorganic materials possess many advantages for drug delivery, like increased loading efficacy, controlled release, stability, and low-toxicity. These properties make them ideal vectors for vaccine delivery. Although various inorganic nanoparticles, such as Au nanoparticles, are extensively studied in vaccine research, the safety issues, such as toxicity, metabolism and side effects, still need to be further evaluated [103,104]. Lipid and protein nanoparticles belong to the organic nanoparticles, which are relatively safe. PLGA is one kind of biocompatible and biodegradable polymer that has been approved by the US Food and Drug Administration (FDA) for human use [105], thus PLGA-based nanoparticles will be a promising platform for vaccine delivery. Besides the platforms, the fabrication process, particle size, dose, characteristics of carried antigens and adjuvants are also important factors that could influence the application of vaccines.

## 5. Conclusions

Taken together, adjuvants participating in innate immunity could initiate innate immune responses and orchestrate the direction and scale of adaptive immune responses. Appropriate administration routes for different adjuvants could differentiate the effects of vaccines because of the uneven distribution of innate sentinel cells. The optimization of combinations of adjuvants is important to regulate the magnitude and breadth of influenza vaccines.

The immune system recognizes many molecules as ligands of innate sensors. The use of the functionally characterized molecules as adjuvants will significantly promote the rational design of influenza vaccines. Simultaneous applications of different adjuvants via particulate carriers are a potential approach to achieve universal influenza vaccines.

## Figures and Tables

**Figure 1 pharmaceutics-13-00068-f001:**
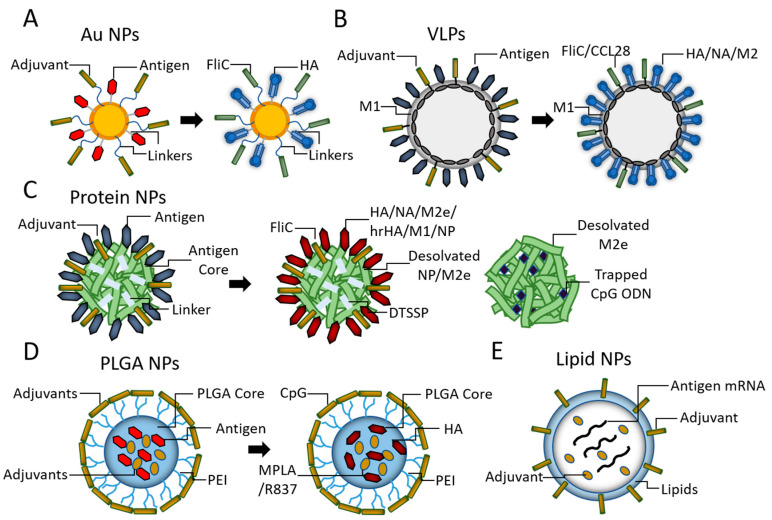
A diagram of different self-adjuvanted nanoparticle (NP) platforms. (**A**) Au NPs. (**B**) virus-like particles (VLP)s. (**C**) Protein NPs. (**D**) biodegradable synthetic polymeric (PLGA) NPs. (**E**) Lipid NPs.

**Table 1 pharmaceutics-13-00068-t001:** Advantages and disadvantages of licensed influenza adjuvants.

Adjuvants	Advantages	Disadvantages
Aluminium Salts	a. Have minor toxicities. b. Improve antigen uptake. c. Increase immune responses.	a. Fail to induce cytotoxic T cell response. b. Ineffective with weak antigens.
Oil-in-water emulsions	a. Induce stronger immune responses including both humoral and cellular immune responses. b. Dose sparing. c.Work efficient with less immunogenic antigens.	a. Highly local reactogenicity. b. Cause systemic symptoms. c. Induce autoimmune disease.
Virosomes	a. Appropriate to wide age groups; b. Facilitate antigen stability, c. Excellent safety profile d. Long-lasting antibody responses	a. Unstable in blood. b. Production and preservation problems.
Heat labile enterotoxin	a. Applicate as mucosal adjuvant	a. Development of Bell’s palsy

**Table 2 pharmaceutics-13-00068-t002:** Potential influenza adjuvants base on immune responses.

TLRs Agonists	TLR3 agonist-poly I:C [25,26,27,28] TLR9 agonist-CpG [38,39,40] TLR4 agonist-MPL, 1Z105 [20,22,24] TLR5 agonist-FliC [29,30,31,32,33,34,35,36,37] TLR7/8 agonist-R837, TRAC-478, 1V270 [20,21,22,23,24]
Cytosolic Nucleic Acids	RLRs receptor agonists: dsRNAs, Small nucleic acids compounds, ploy I:C [41,42,43] STING agonist-cGAMP [45,46,47,48]
Inflammasomes Agonists	NLRC4 inflammasome-FliC [55] NLRP3 or AIM2 inflammasomes-Nucleic acids (DNA and RNA) [56,57,58]
Immune Cells Activator	iNKT cells activator- α-GalCer [63] Mast cells activator-C48/80, IL-33, IL-18 [64,65,66,67]
Cytokines and Chemokines	IL-1β [68,69] TNF [65,67] GM-CSF, GIFT4 [71,72,73] CCL27, CCL28 [74,75]

Abbreviations: TLRs, Toll-like receptors; RLRs, RIG-I-like receptors; STING, Stimulator of interferon genes; cGAMP, 2’3’-cyclic GMP-AMP; NLRC4, NLR-family CARD domain-containing protein 4; NLRP3, NOD-like receptor protein 3; iNKT cells, Invariant (i) natural killer T cells; α-GalCer, Glycolipid ligand α-galactosylceramide; TNF, Tumor necrosis factor; GM-CSF, Granulocyte-macrophage colony-stimulating factor.

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
