# Peer review of "Promising Adjuvants and Platforms for Influenza Vaccine Development"

_pharmaceutics, 2021, doi:10.3390/pharmaceutics13010068_

Round 1
Reviewer 1 Report
Authors reviewed the ample of adjuvants being used and evaluated in influenza vaccines. The authoritative review reflects the author’s expertise in the field. Authors have nicely discussed their own experiences in the influenza vaccine development, particularly nanotechnology-based vaccine adjuvant development.
Major suggestions
Although the descriptive text is providing the concept of novel influenza vaccine adjuvants, diagrammatic illustrations are missing, which would bring more elegance to the manuscript.
Some minor suggestions
- Line 31: Because humans have no immunity to an emerging pandemic virus, an influenza ---à As a development of immunity an emerging pandemic virus takes time, an influenza
- Line 64: Describe the composition of MF59, AS03, and AF03.
- Line 66: roles, such --à ways, such
- Line 88: dendritic cells (DCs), -à DCs
- Line 91: oligomerization domain (NOD)-like receptors -----à Nucleotide-binding and oligomerization domain (NOD)-like receptors
- Section 3: Referencing one more publication would give more clarity… https://doi.org/10.1016/j.tips.2017.06.002.
- Table 1 requires footnotes with abbreviations.
- Use of abbreviations is not proper throughout the manuscript…few abbreviations are repeated, e.g., DCs, GC; few acronyms are used without abbreviating, e.g., VLPs; few are not et al abbreviated, e.g., MNP, BALT. Please correct these minor issues.
Author Response
Reviewer1
Comments and Suggestions for Authors
Authors reviewed the ample of adjuvants being used and evaluated in influenza vaccines. The authoritative review reflects the author’s expertise in the field. Authors have nicely discussed their own experiences in the influenza vaccine development, particularly nanotechnology-based vaccine adjuvant development.
Major suggestions
Q: Although the descriptive text is providing the concept of novel influenza vaccine adjuvants, diagrammatic illustrations are missing, which would bring more elegance to the manuscript.
A: One diagram of different self-adjuvanted nanoparticle platforms is added in Section 4 according to the suggestion.
Some minor suggestions
Q1: Line 31: Because humans have no immunity to an emerging pandemic virus, an influenza ---à As a development of immunity an emerging pandemic virus takes time, an influenza
A1: Change has been made (Highlight in line 31).
Q2: Line 64: Describe the composition of MF59, AS03, and AF03.
A2: The composition has been added (Highlighted in lines 65-66).
Q3: Line 66: roles, such --à ways, such
A3: Change has been made (Highlighted in line 67).
Q4: Line 88: dendritic cells (DCs), -à DCs
A4: Change has been made (Highlighted in line 91).
Q5: Line 91: oligomerization domain (NOD)-like receptors -----à Nucleotide-binding and oligomerization domain (NOD)-like receptors
A5: Change has been made (Highlighted in line 94).
Q6: Section 3: Referencing one more publication would give more clarity… https://doi.org/10.1016/j.tips.2017.06.002.
A6: It has been cited (Highlighted in line 90).
Q7: Table 1 requires footnotes with abbreviations.
A7: The footnotes with abbreviations have been added.
Q8: Use of abbreviations is not proper throughout the manuscript…few abbreviations are repeated, e.g., DCs, GC; few acronyms are used without abbreviating, e.g., VLPs; few are not et al abbreviated, e.g., MNP, BALT. Please correct these minor issues.
A8: Changes have been made and highlighted. (DCs in lines 91 and 191; GC in lines 214 and 274; VLPs in lines 127 and 221; MNP in line 296; BALT in line 304).

Reviewer 2 Report
Wang et al. summarized a timely mini-review for adjuvants and platforms for influenza vaccine. It is a topic of importance. A few minor improvements are expected before the acceptance of the manuscript.
- The basic concept/definition of "adjuvant" is missing at the introduction section. A few more sentences are needed to discuss about what they are.
- The whole manuscript has zero figure. An extensive supplement of figures is needed for most of the sections.
- What are the advantages, safety concerns, technical barriers that existing for the platforms mentioned in Section 4? Summarizing these information as a Table would be helpful.
- Why are these unnatural synthetic platforms (e.g. polymers, inorganic nanoparticles) needed for adjuvant development?
Author Response
Reviewer 2
Comments and Suggestions for Authors
Wang et al. summarized a timely mini-review for adjuvants and platforms for influenza vaccine. It is a topic of importance. A few minor improvements are expected before the acceptance of the manuscript.
Q1: The basic concept/definition of "adjuvant" is missing at the introduction section. A few more sentences are needed to discuss about what they are.
A1: It has been added in lines 51-52 in the introduction section.
Q2: The whole manuscript has zero figure. An extensive supplement of figures is needed for most of the sections.
A2: One Table (Table 2) is included in this manuscript to display all the adjuvants mentioned in Section 3. We have added another Table (Table 1) in Section 2 to show the advantages and disadvantages of the licensed influenza adjuvants. One more figure (Figure 1) has been added in Section 4 according to the suggestion to display different self-adjuvanted nanoparticle platforms.
Q3: What are the advantages, safety concerns, technical barriers that existing for the platforms mentioned in Section 4? Summarizing these information as a Table would be helpful.
A3: We have added one paragraph to briefly discuss the safety of nanoparticle platforms (Highlighted in lines 336-345).
Q4: Why are these unnatural synthetic platforms (e.g. polymers, inorganic nanoparticles) needed for adjuvant development?
A4: We summarize different vaccine and adjuvants delivery platforms in this review including the polymer and inorganic nanoparticles because they are also an important part of vaccine delivery systems. We inserted the advantages of the polymer and inorganic nanoplatforms for vaccine delivery in lines 336-343.
Thanks.

Reviewer 3 Report
The review manuscript entitled “Promising adjuvants and platforms for influenza vaccine development” is covered new information on the updated developments. But, it needs some of the edits.
First of all, differenr types of review articles has been published on the influenza vaccine development. Please explain the importance of the current review in comparison with some of the below published articles.
Previously, Kumar A, Meldgaard TS, Bertholet S. Novel platforms for the development of a universal influenza vaccine. Frontiers in immunology. 2018 Mar 23;9:600 has been published the updated delivery systems of influenza. What is the additional information covered in the present manuscript.
Jazayeri SD, Poh CL. Development of universal influenza vaccines targeting conserved viral proteins. Vaccines. 2019 Dec;7(4):169. This particular review covers the viral proteins as main targeted. Add a detailed difference of the present review with respect to this one in the manuscript.
Wei CJ, Crank MC, Shiver J, Graham BS, Mascola JR, Nabel GJ. Next-generation influenza vaccines: opportunities and challenges. Nature Reviews Drug Discovery. 2020 Feb 14:1-4. This is also latest published review on the same topic.
Further, the author also published some part of the review recently in Virus MDPI journal
Tang S, Zhu W, Wang BZ. Influenza Vaccines toward Universality through Nanoplatforms and Given by Microneedle Patches. Viruses. 2020 Nov;12(11):1212.
Deng L, Wang BZ. A perspective on nanoparticle universal influenza vaccines. ACS infectious diseases. 2018 Nov 5;4(12):1656-65.
Author Response
Reviewer 3
Comments and Suggestions for Authors
The review manuscript entitled “Promising adjuvants and platforms for influenza vaccine development” is covered new information on the updated developments. But, it needs some of the edits.
First of all, differenr types of review articles has been published on the influenza vaccine development. Please explain the importance of the current review in comparison with some of the below published articles.
Q1: Previously, Kumar A, Meldgaard TS, Bertholet S. Novel platforms for the development of a universal influenza vaccine. Frontiers in immunology. 2018 Mar 23;9:600 has been published the updated delivery systems of influenza. What is the additional information covered in the present manuscript.
A1: Compared to this review, we focus on different aspects. We discussed innate immune responses triggered by adjuvants in Section 3. Particulate adjuvants and self-adjuvanted particulate vaccine platforms were comprehensively described in Section 4.
Q2: Jazayeri SD, Poh CL. Development of universal influenza vaccines targeting conserved viral proteins. Vaccines. 2019 Dec;7(4):169. This particular review covers the viral proteins as main targeted. Add a detailed difference of the present review with respect to this one in the manuscript.
A2: Compared to this review, we focused on the promising adjuvants and platforms which are different aspects of the development of influenza vaccines.
Q3: Wei CJ, Crank MC, Shiver J, Graham BS, Mascola JR, Nabel GJ. Next-generation influenza vaccines: opportunities and challenges. Nature Reviews Drug Discovery. 2020 Feb 14:1-4. This is also latest published review on the same topic.
A3: This review provided some latest information about the influenza vaccines. However, we focused on adjuvants and platforms that are less mentioned in this review.
Q4: Further, the author also published some part of the review recently in Virus MDPI journal
Tang S, Zhu W, Wang BZ. Influenza Vaccines toward Universality through Nanoplatforms and Given by Microneedle Patches. Viruses. 2020 Nov;12(11):1212.
Deng L, Wang BZ. A perspective on nanoparticle universal influenza vaccines. ACS infectious diseases. 2018 Nov 5;4(12):1656-65.
A4: Thank you for your question. We have published several reviews previously related to the development of universal influenza vaccines. Two of the reviews mentioned above were focusing on structure-based antigen design/construction, nanoparticle platforms, and novel delivery methods. These are two directions of our influenza vaccine researches. And in this review, we emphasize the promising adjuvants and their delivery strategies in the formulation of influenza vaccine.
Thanks.
Round 2
Reviewer 2 Report
Comments from previous reviewers have been properly addressed.
Reviewer 3 Report
Agree with author responses.